# Diversity and complexity of arthropod references in haiku

Andrew R. Deans [ID][◑]*, Laura Porturas[◑]

Frost Entomological Museum, The Pennsylvania State University, University Park, Pennsylvania, United States of America

◑ These authors contributed equally to this work.
* adeans@psu.edu

**Data Availability Statement:** We provide here those poems that are in the public domain and those we have permission to share. Uniform Resource Identifiers (URIs) and International Standard Book Numbers (ISBNs) are provided to the source publications, which allow anyone with

## Abstract

Haiku are short poems, each composed of about 10 words, that typically describe moments in nature. People have written haiku since at least the 17th century, and the medium continues to be popular with poets, amateurs, educators, and students. Collectively, these poems represent an opportunity to understand which aspects of nature—e.g., which taxa and biological traits—resonate with humans and whether there are temporal trends in their representation or the emotions associated with these moments. We tested this potential using a mix of linguistic and biological methods, in analyses of nearly 4,000 haiku that reference arthropods. We documented the taxa and the life history traits represented in these poems and how they changed over time. We also analyzed the poems for emotion and tone. Our results reveal a mix of predictable trends and compelling surprises, each of which stand to potentially inform engagement strategies. At least 99 families of arthropods, in 28 orders, are represented in these haiku. The eight most commonly referenced taxa, from highest to lowest number of references, include: Lepidoptera, Hymenoptera, Diptera, Coleoptera, Araneae, Orthoptera, Hemiptera, and Odonata. Several common, conspicuous orders were never referenced, including Trichoptera, Plecoptera, and Megaloptera. The most commonly referenced traits relate to ecology (especially habitat, phenology, time of day), behavior (especially sound production), phenotype (especially color), and locomotion (especially flight). The least common traits in haiku relate to arthropod reproduction and physiology. Our analyses revealed few obvious temporal trends in the representations of taxa, biological traits, or emotion and tone. The broader implications of these results and possible future directions are discussed.

## Introduction

Life abounds with ephemeral moments that evoke emotion, expose the profound, or otherwise effect inspiration. Imagine a dragonfly gliding effortlessly across a pond, a millipede slowly coiling in self defense, or a caterpillar smoothly descending from the canopy on a line of silk. Each of these seemingly simple events has the potential to awaken or reinforce one's connection to the natural world. People often capture these experiences in short poems, called *haiku*.

access to the Internet to reconstruct the full corpus. All of the digital sources for the poems analyzed in this study are open access. The data sets provided here do have all all the data used in our statistical analyses and those data that were used to generate figures.

**Funding:** The author(s) received no specific funding for this work.

**Competing interests:** The authors have declared that no competing interests exist.

A haiku conveys such a moment intentionally and objectively, usually in about ten words, leaving the reader to infer emotional and sensory elements of that moment [1].

The literary origins of haiku date to the 17th century in Japan [2], or even earlier [3], but the popularity of haiku has soared globally over the last few decades [4]. There are now dozens of societies, journals, and competitions dedicated to the craft. The medium is also employed in countless classrooms worldwide, in exercises that teach mindfulness, grammar, and effective communication, among other educational and therapeutic goals [5]. Traditionally, these poems focused on natural events, and each haiku included a word or phrase, referred to as *kigo* [6], that alluded to the season [2]. Consider this haiku, by Mastuo Bashō (1644–1694), who is broadly considered to be the original haiku master [7]:

Spring rain falling

The roof leaks,

Trickling down the wasps' nest.

"Spring rain" serves as the kigo. The reader also witnesses the wasps' nest dripping with rain and can imagine the unstated elements Bashō experiences in that moment.

Thousands of haiku are published annually, and certainly tens of thousands or more are written each year. Collectively, they offer opportunities to understand how and which organisms inspire people, which biological traits resonate with the human experience, and where openings may exist for more effective outreach regarding biodiversity and conservation [8, 9]. Given the long history of haiku, one could also examine how biological references in haiku have changed temporally. Poets may incorporate increasingly sophisticated references over time, for example, as shared knowledge of organismal biology increases. Alternatively, the diversity of biology represented in haiku may decrease over time, as people spend more time indoors or otherwise have fewer encounters with wildlife [10–12].

We aim here to explore this potential, by analyzing a set of haiku, broadly defined, using linguistic and biological approaches. Our focus is on haiku that reference insects and their relatives, including arachnids and myriapods (Arthropoda). Here, we address the following questions: (1) What arthropods and which aspects of their biology are represented in haiku? (2) Have arthropod references in haiku changed over time? (3) What can we glean from the language used in haiku, regarding public sentiment towards different arthropod taxa and their associated biological traits?

## Materials and methods

### Corpus assembly

We assembled a primary corpus of 3,894 relevant haiku, internationally sourced and written by 1,248 individual poets between and 1549 and 2022. These poems were acquired from more than 65 unique sources (>400 volumes), including haiku-focused websites, journal volumes, books, and poetry competitions. About one third of the poems (n = 1129) were sourced from a poetry contest specifically focused on insect-related haiku, the Hexapod Haiku Challenge (HHC; [13]). This source includes 236 haiku written by poets under the age of 13, which provides a small window into potential age-related differences in arthropod references. Only haiku that were written in English or had been translated into English were included in our analysis. A complete list of sources is available in S1 Appendix. The haiku were extracted manually from each source and organized in a spreadsheet with relevant metadata, using Dublin

Core [14] terms: "contributor" (who added the poem to the data set), "creator" (poet's name), "date" (date of publication or poet's death year, if publication year unknown), "dateSubmitted" (timestamp for when the poem was added to the data set), and "source" (the resource from which the haiku was found). Additional headers for "poem" (the text of the poem itself, pasted without regard to formatting), "identifier" (unique string for reference), and "notes" (any comments made by the contributor) were also used. The primary corpus is available as S2 File, which includes the poems we have permission to share and URIs to those haiku still under copyright.

## Scoring biological complexity

A subset of haiku was compiled as a secondary corpus and scored for 69 variables related to biological complexity and the lowest-level taxon represented. Given the labor intensive nature of this process, we limited our scoring to 2,500 haiku S1 File, representing each creator in the primary corpus and including 1,003 HHC poems. Detailed descriptions of our scoring method and variables are provided in supplementary document S2 Appendix. A list of the variable categories and two scoring examples are provided below. To calculate a haiku's complexity, we scored variables as present (+1) or absent (0) for each major category below and for each of its lower-level variables:

- **locomotion**: ambulatory, cursory, saltatory, natatory, skating, flight, in place

- **reproduction**: courtship, mating, oviposition, hatching, brood care

- **anatomy/phenotype**: legs, antenna, mouthparts, head, wings, genitalia, defensive structure, setae, color/pattern, extended phenotype

- **physiology**: digestion, molting, metamorphosis, vision, chemosensory, hearing, neural, thermoregulation, bioluminescence, phototaxis, biological sex

- **life stage**: egg, immature, pupa, imago (adult)

- **behavior (not reproduction)**: mimicry, predation, prey, aposematism, parasitism, foraging, pollination, feeding, defense, sound production, sociality, migration, dispersal, grooming

- **ecology**: domestic, peridomestic, not domestic (wild), association, paleontological, phenological (seasonal), temporal (circadian)

As an example, consider this haiku from 1812, penned by Kobayashi Issa and translated by David G. Lanoue [15]:

the hairy bug

becomes a butterfly . . .

summer moon

We scored it as positive (+1) for setae ("hairy"), which is a variable of anatomy/phenotype (+1); for metamorphosis ("becomes"; +1), which is a variable of physiology (+1); for adult (+1) and larva (+1); for the explicit phenology ("summer"; +1) and allusive temporal ("moon"; +1) references, which are variables of ecology (+1). This matrix yielded a complexity score of 13.04 (= 9/69 × 100) for this poem. Based on the word "butterfly", the lowest-level taxon was recorded as Papilionoidea. Another poem by Issa, from 1806 [15], scored considerably lower in biological complexity:

while swatting a fly

today again . . .

the mountain temple bell

This poem vaguely references locomotion (people swat at flies that are either flying or in place; +1) and ecology ("mountain temple" hints at the fly's habitat; +1), and it references life stage (imago; +1). This matrix yields a complexity score of 4.35 (= $3/69 \times 100$). Based on the word "fly", we recorded the lowest-level taxon as Diptera.

## Linguistic analyses

We analyzed the secondary corpus for word use, tone, and emotion using the Linguistic Inquiry and Word Count (LIWC) application (LIWC-22; [16]). LIWC-22 compares contents of the corpus with built-in word banks associated with each linguistic category. Output scores for the trait categories from this analysis were appended to our dataset. We focused on the following linguistic variables: positive tone (tone_pos), negative tone (tone_neg), emotion (emotion), positive emotion (emo_pos), negative emotion (emo_neg), and anxiety (emo_anx). These scores are the proportion of words in the haiku that match to the LIWC software word-banks for each of the different linguistic categories/variables. If a haiku has 9 words total and 3 of them match words in the tone_pos word bank, the haiku would have a score of (3/9 =) 0.333 for that variable. If it also has 1 word that matches words in the tone_neg word bank, it receives a (1/9 =) 0.111 for that variable.

Word clusters were identified using AntConc [17, 18] on the primary corpus, focusing especially on the most common arthropod words (with wildcards): butterfl*; spider*||web; ant|| ants; firefl*; dragonfl*; crick*; cicad*; mosquito*; moth*; etc. More details regarding AntConc settings, variables, and early results can be found in supplementary document S3 Appendix. The AntConc analyses were used to highlight linguistic trends, references to each taxon over time, and reveal potentially overlooked connections between taxa and biological traits.

## Taxon hierarchies and diversity

To more fully understand the taxonomic richness of our dataset and to explore broader taxonomic trends, we placed each reference of an arthropod (i.e., each lowest-level taxon) within a taxonomic hierarchy. Some haiku referenced more than one taxon, and each reference was separated as an independent record. Higher-level taxonomic names were obtained for referenced taxa using the "tax_name" function in the package "taxize" [19], in R Statistical Software version 4.2.1 [20]. We chose to query only the National Center for Biotechnology Information (NCBI) [21] database for this step. Deviations from this query include the following: (a) we use Anthophila to represent bees, rather than the broader taxon Apoidea, and (b) we treat Psocodea as an order, in line with our current understanding of insect classification.

To get a better sense of whether each taxon was over- or underrepresented in haiku, relative to its diversity and likelihood of human encounter, we compared the number of references to their alpha diversity (number of species; see references in Discussion) and the proportion of arthropod occurrences on iNaturalist [22]. iNaturalist is a Web-based application, where people can upload records, including photos and sound recordings, of organisms they encounter. Unlike other, similar resources, for example the photo sharing application Flickr (https:// flickr.com/), iNaturalist uses a taxonomic hierarchy to organize their data, which allowed us to more readily estimate rates of encounter.

### Recording haiku source

Haiku that were sourced from the Hexapod Haiku Challenge were categorized as "HH", and the other haiku collected websites, journal volumes, books, and poetry competitions were categorized as "other".

### Statistics

A generalized linear model (family = quasipoisson) was used to examine whether time, haiku source, or # of authors was a predictor for taxon diversity (estimated by # of unique taxonomic orders/year) referenced in haiku. The analysis was restricted to years that had 10 or more haiku, and performed using the glm() function in base R (R version 4.2.1) [20].

A linear mixed effects model was used to examine whether time, haiku source, or # of authors was a predictor for biological complexity referenced in haiku. Time, haiku source, and # of authors were included as fixed effects, and taxonomic order was included as a random effect. The analysis was restricted to years that had 10 or more haiku, and performed using the function lmer() from the package "lme4: Linear Mixed-Effects Models using 'Eigen' and S4" [23].

Nonparametric multivariate analyses were used to determine whether any of the LIWC variables related to the tone & emotion of haiku in the dataset (tone_pos, tone_neg, emotion, emo_pos, emo_neg, emo_anx) were dependent on categorical groups of interest (e.g., haiku written about adult insects vs. immature insects, or domestic vs. outdoor arthropods). We used the function nonpartest() from the package "npmv: Nonparametric Comparison of Multivariate Samples" [24]. When all four test statistics can be calculated (ANOVA-Type, Lawley-Hotelling Type, Bartlett-Nanda-Pillai-Type, Wilks' Λ-Type), it is recommended the Wilks' Λ-Type statistic be used [25]. For each comparison, permutation replications were set to 1000, and prior to analysis, haiku that referenced more than one of the categorical variables were removed (e.g., a haiku that referenced both adult insects and immature insects would be excluded) to help eliminate the possibility of response values being incorrectly applied to multiple categories.

To summarize the strength and direction of the relationships between LIWC variables and time (date), we calculated Spearman's Rank correlation coefficient using the cor.test() function in base R (R version 4.2.1) [20].

The R packages "dplyr: A Grammar of Data Manipulation" [26] and "tidyr: Tidy Messy Data" [27] were both used to manipulate and organize data. the R package "ggplot2: Create Elegant Data Visualisations Using the Grammar of Graphics" [28] were used to generate some figures.

## Results

### Taxonomic trends

The 2,500 haiku in our secondary corpus included 2,611 references to arthropods, with reference to 24 of the 33 major taxonomic groups in Fig 1 (far right column). The referenced arthropods come from 28 different taxonomic orders and represent at least 99 taxonomic families. Some taxa are predictably well-represented (Fig 2). Lepidoptera, for example, which includes many large, colorful, and otherwise conspicuous species of butterfly and moth, had the most references (n = 510). Hymenoptera (ants, wasps, bees; n = 347), Diptera (mosquitoes, flies; n = 314), Coleoptera (fireflies, beetles; n = 310), and Araneae (spiders; n = 220) round out the top five most referenced taxa. Each of these taxa includes well over 50,000 named species worldwide, many of which are conspicuous and familiar.

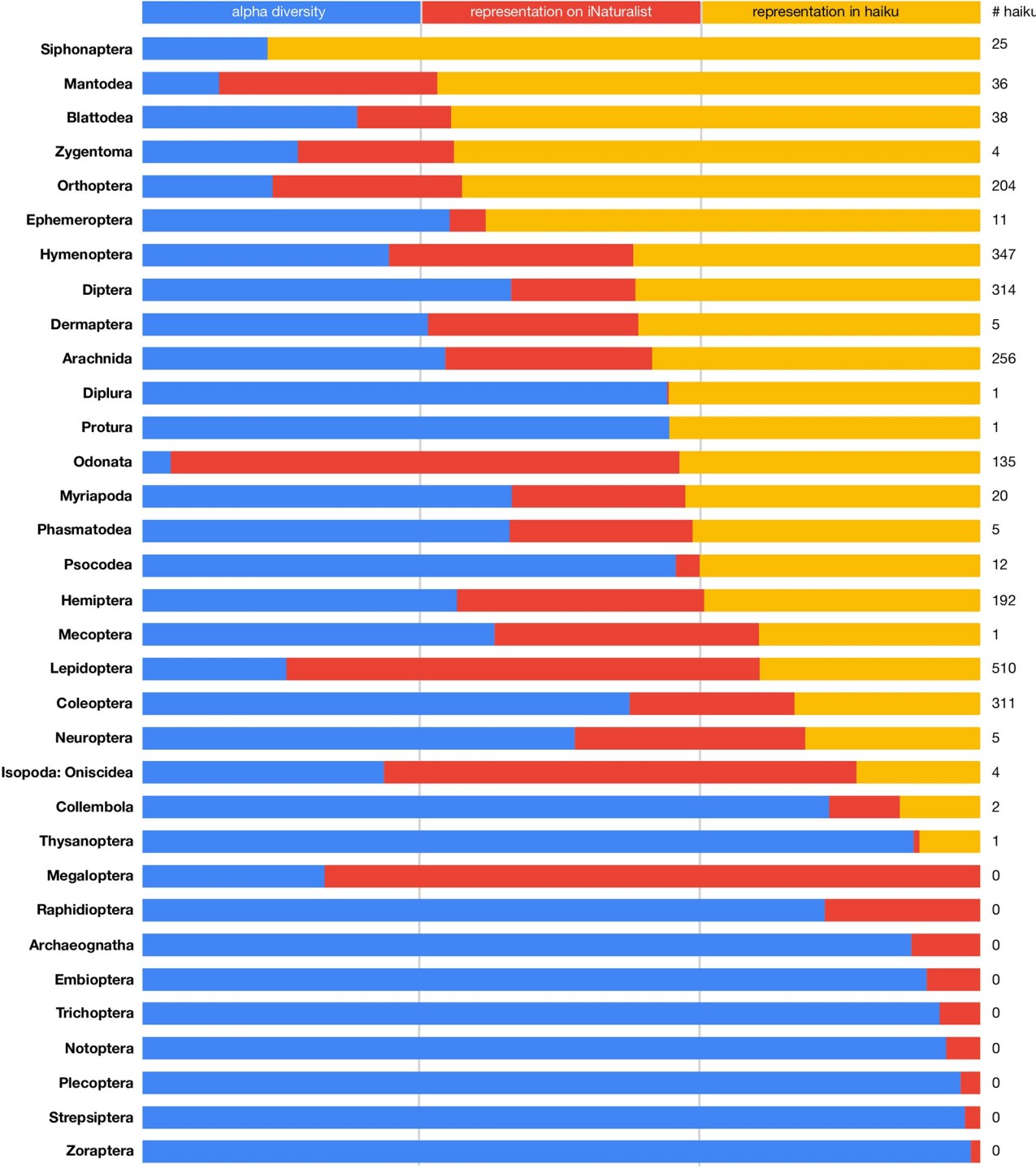

**Fig 1. Relative arthropod representation.** Bar size represents proportion in each category, relative to other categories. Blue = alpha diversity (number of species); red = representation on iNaturalist; yellow = representation in haiku. Three bars of equal size for a taxon would indicate that its representation is proportional across all categories (i.e., that the taxon comprises the same percentage in each category). See Discussion for references regarding the diversity of each taxon.

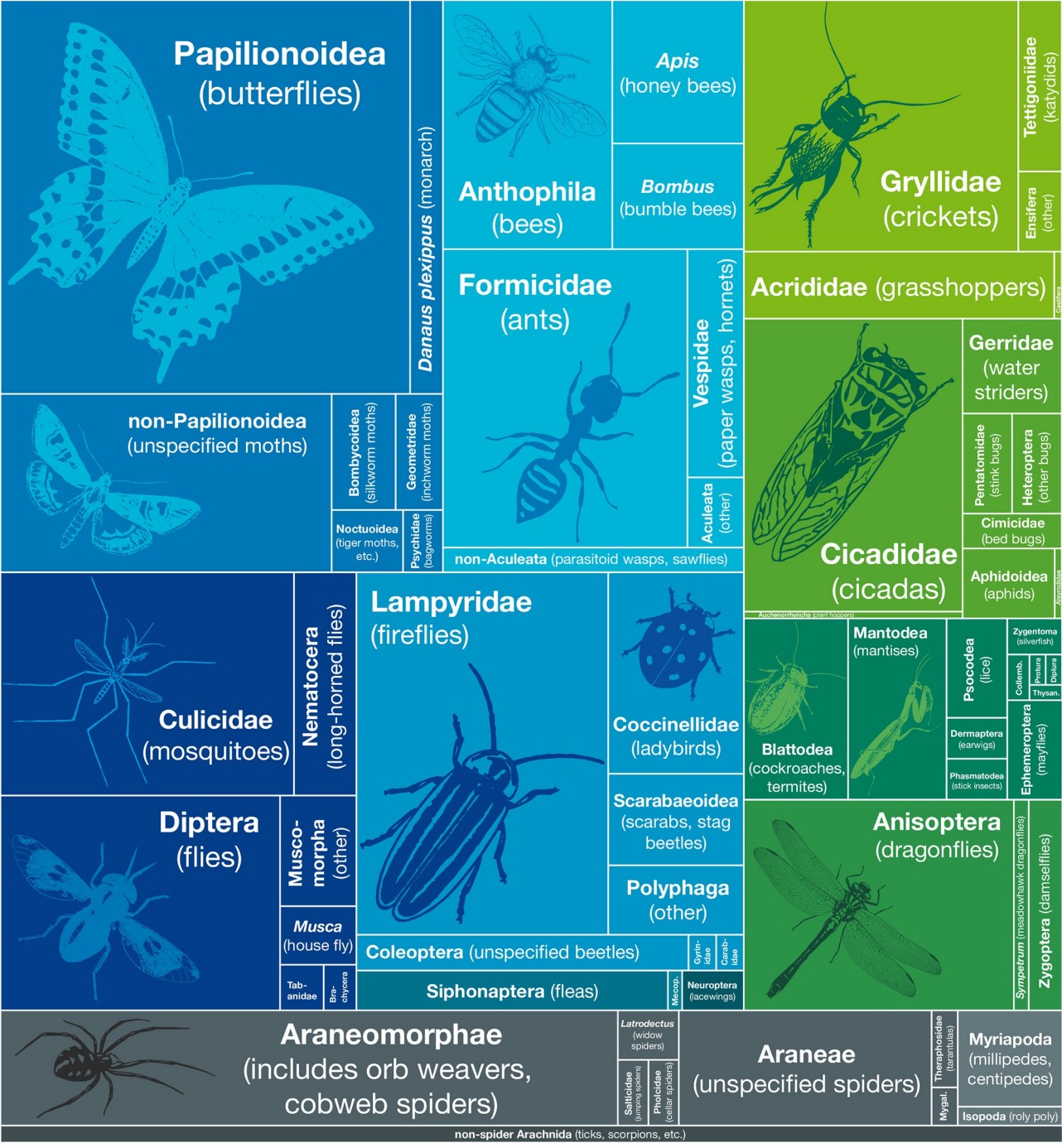

**Fig 2. Taxon representation in haiku.** Rectangle sizes are proportional and represent major taxonomic names referenced in haiku. The arrangement is roughly phylogenetic, with blue rectangles representing Holometabola, green rectangles non-holometabolous Hexapoda, and gray representing non-hexapod Arthropoda.

Nine higher-level taxa were not referenced in any haiku (Fig 1, bottom rows), and four were only referenced once. Predictably, some of these taxa are depauperate and/or tend to have species of very small size. Zoraptera (zero references), for example, are small, cryptic insects with stringent habitat requirements [29] and <50 species. Protura (one reference) are even smaller

in size, generally requiring a microscope to observe. Supplementary S1 File contains the raw numbers for representation by taxon, which are summarized by Fig 2 and S1 Table.

As described above, we compared the taxonomic representation against alpha diversity and proportion of arthropod occurrences on iNaturalist, to get a better sense of whether each taxon was over- or underrepresented in haiku, relative to its diversity and likelihood of human encounter. Lepidoptera is the most frequently referenced taxon in our corpus, for example, accounting for 20.9% of arthropod references (yellow bar in Fig 1). Lepidoptera is overrepresented in haiku, relative to the number of species in the taxon (13.6% of all arthropod species; blue bar in Fig 1) but underrepresented relative to their frequency of encounter on iNaturalist (44.95% of the almost 14,000,000 arthropod observations recorded on that platform; red bar in Fig 1). The representation of Hemiptera in haiku (7.87% of references) closely resembles their species diversity (8.95% of all arthropod species) and likelihood of encounter (7.06% of arthropod occurrences on iNaturalist). Siphonaptera (fleas), on the other hand, are vastly overrepresented in haiku (1.02% of arthropod references), relative to their species-level diversity (2,075 species; 0.18% of arthropods) and frequency of encounter on iNaturalist (n = 73; 0.0005% of occurrences). These trends are discussed further in the taxon-specific summaries below, which also includes the results related to subordinal taxa.

Examining the relative proportions of taxa, binned by decade (S1 Fig) also reveals several notable trends. References to fleas (Siphonaptera), for example, are much more abundant in haiku prior to 1900, possibly because of improvements in pest control [30], hygiene related to livestock/pets, and increased urbanization [31]. References to bed bugs (Hemiptera: Cimicidae), however, are all contemporary (2010–2022), likely due to their recent resurgence globally [32, 33]. References to a third group of common, flightless parasites, sucking lice (Psocodea: Anoplura), are relatively evenly distributed across time. The relative consistency of taxonomic representation since 1950 (S1 Fig) is also notable. However, we cannot discount the fact that our uneven temporal sampling of haiku, or even mistranslation into English, may explain or at least influence some of these trends. Subordinal trends are presented and discussed in the taxon summaries below.

We determined that there were significant positive effects of both time and number of authors on the taxonomic diversity referenced in haiku ($t(33) = 4.09$, $p<.001$ and $t(33) = 3.63$, $p<.001$, respectively), and a significant negative effect of haiku source "other" $t(33) = -2.23$, $p = 0.026$).

## Biological trends

The insect biology referenced in these 2,500 haiku was similarly sophisticated yet uneven (Fig 3). The average haiku, composed of merely nine or ten words, had a complexity score of 7.52, meaning that just over five out of the 69 traits we scored (three major trait categories plus two more specific traits) were referenced. The poem with the highest complexity score (20.29), written by Gina Burns in 2010, referenced 14 different biological traits (9 specific traits in 5 major categories):

Tiny eggs on glass

Like jeweled beads for our eyes

Hatch, fly, eat, mate, lay

Seventeen haiku scored as zero for biological complexity, for example this haiku by John Soules [34].

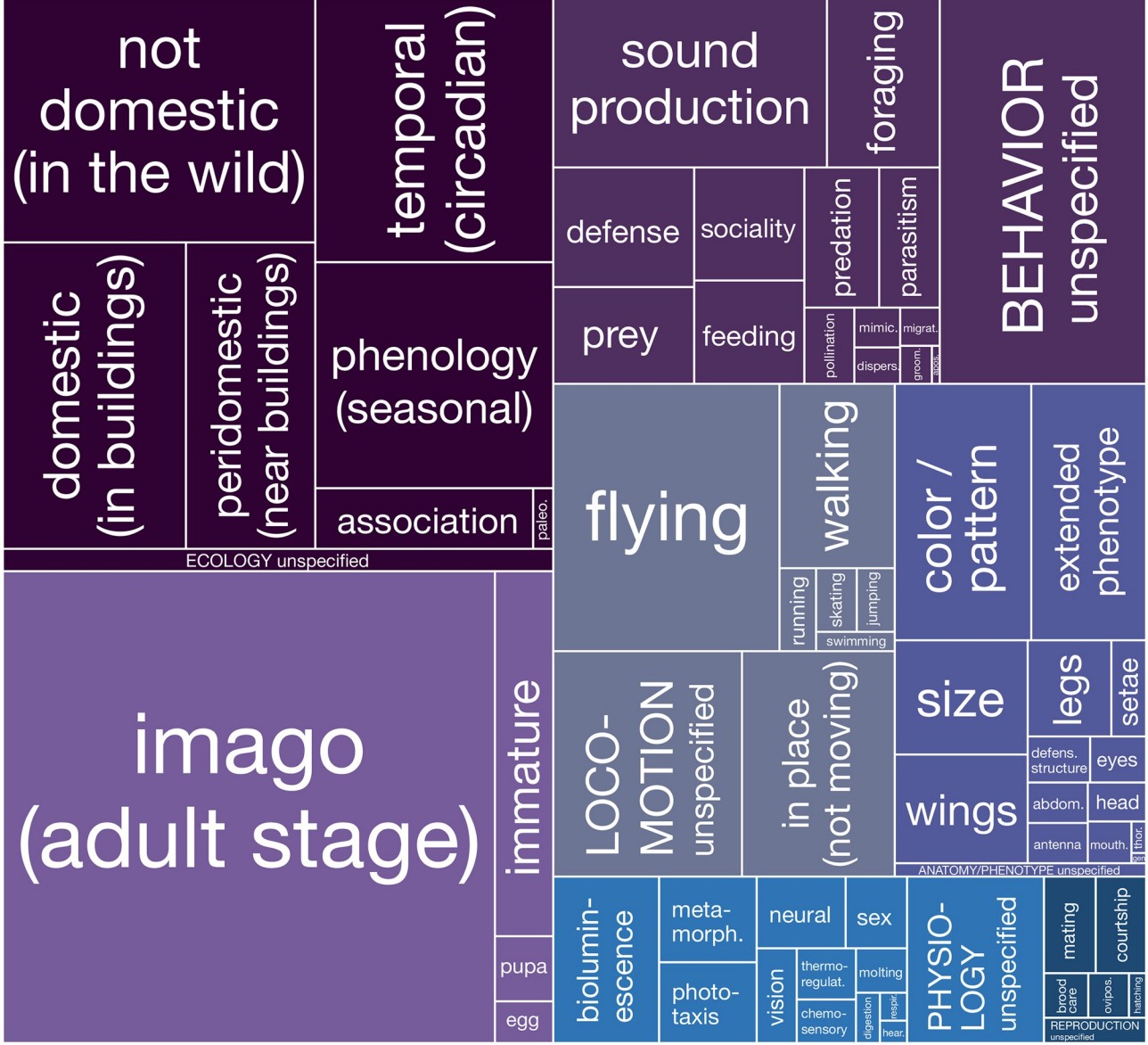

**Fig 3. Biological traits referenced in haiku.** Rectangle sizes are proportional to the number of references for each category. Labels in ALL CAPS refer to subtle references to major biological categories that could not be confidently assigned to any one subcategory.

> mayfly—
>
> all the promises
>
> we made

The arthropods' relationships with their environment, what we refer to as their "ecology", were by far the most commonly referenced traits. More than 83% of the haiku in our corpus (n = 2,082) referred to ecology, and five of the top 10 most referenced biological traits fit this

theme: habitat (not domestic (wild), n = 506; domestic, n = 369; peridomestic, n = 261), time of day (temporal, n = 416), and time of year (phenology; n = 354). Other variables in the top 10 relate to life stage (adult, n = 1,529), locomotion (flying, n = 398; in place, n = 227), behavior (sound production, n = 308), and phenotype (color/pattern, n = 230; extended phenotype, n = 194). By contrast, only 112 haiku (<5%) refer to aspects of reproduction, the least popular category of biological references.

To examine temporal trends in the representation of different biological traits in haiku, we plotted the relative proportions of the top 20 biological traits (S3 Fig) and the major trait categories, binned by decade (S2 Fig). Unsurprisingly, the biological traits and the larger biological categories show similar trends through time. Of note, the relative proportions of different biological traits referenced remain fairly stable through time, with the exception of those decades with small sample sizes.

We found that neither time, number of authors, nor haiku source influenced the complexity of biological traits references (t(332) = 1.90, p = 0.058, t(332) = 0.23, p = 0.819, and t(332) = 0.91, p = 0.36, respectively).

## Tone and emotional language trends

The following comparisons test whether the defined categories influence the language related tone and emotion in haiku.

**Adult vs. immature arthropods.** For this comparison, only the ANOVA-Type statistic could be computed, and we found no evidence that the language related to tone and emotion differs between the haiku referencing adult (n = 1500) vs. immature (n = 111) arthropods (ANOVA-Type statistic$_{3.056,1252.642}$ = 1.043, p = 0.373, permutation p = 0.369).

**Domestic vs. outdoor arthropods.** Despite the fact that increasing urbanization and encounters with insects indoors are correlated with more negative opinion of insects [35, 36], we found no evidence that language differs between the haiku referencing domestic (n = 367), peridomestic (n = 259), or not-domestic (n = 506) (Wilks' Λ-Type statistic$_{12,2248}$ = 1.119, p = 0.340; permutation p = 0.082).

**Between taxonomic orders.** Only orders with 10 or more references were included. Our results reveal there are differences in the language used between taxonomic orders (Araneae n = 227, Blattodea = 38, Coleoptera = 311, Diptera = 314, Ephemeroptera = 11, Hemiptera = 192, Hymenoptera = 347, Lepidoptera = 510, Mantodea = 35, Odonata = 135, Orthoptera = 204, Psocodea = 12, Scorpiones = 15, Siphonaptera = 25) (Wilks' Λ-Type statistic$_{78.000,13001.710}$ = 2.901, p<0.001; permutation p<0.001). The relative effects (probability that a random sample from one group is higher than a random sample from another group) of taxon order on the LIWC variables indicating attitude were strongest in haiku referencing Scorpiones (scorpions); relative effects for negative tone = 0.80627, negative emotion = 0.73256, anxiety = 0.62604, and emotion = 0.70240. Haiku referencing scorpions also had a lower relative effect regarding positive tone = 0.43248. Haiku referencing Psocodea (lice) had more subtly elevated relative effects for negative tone = 0.58240, negative emotion = 0.55433, and emotion = 0.57172, but surprisingly also had an elevated effect for positive tone = 0.58218. Haiku referencing Ephemeroptera (mayflies) had a slightly elevated relative effect for positive tone = 0.55293, and a low relative effect regarding negative tone = 0.42361. Surprisingly, haiku referencing Mantodea (mantids) had an elevated negative tone = 0.56744. The remaining relative effects of taxon order on the LIWC variables ranged between 0.45 and 0.55, which are very close to 0.50, indicating little to no probability of differences in language that reflect attitude.

**Flying vs. non-flying other types of locomotion.** For this comparison, only the ANOVA-Type statistic could be computed, and we found no evidence that language use differs

between the haiku referencing flying insects (n = 376) vs. arthropods using other types of locomotion (n = 404) (ANOVA-Type statistic$_{3.072,2386.679}$ = 0.958, p = 0.413; permutation p = 0.41).

**Singing vs. non-singing.** For this comparison, only the ANOVA-Type statistic could be computed, and we found evidence for differences in language used between haiku referencing singing insects (n = 308) vs. haiku that did not reference singing (n = 2192) (ANOVA-Type statistic$_{2.92,3145.172}$ = 8.49, p<0.001, permutation p<0.001). However, the effects were very modest, and the haiku written about singing insects were slightly more likely to have both positive and negative language in the haiku (relative effects: positive tone = 0.53683, negative tone = 0.53377, negative emotion = 0.53457, and emotion = 0.53869).

**Color vs. other.** For this comparison, only the ANOVA-Type statistic could be computed, and we did find that there were significant differences between haiku referencing insect coloration (n = 230) vs. those that did not (n = 2270) (ANOVA-Type statistic$_{2.938,2444.1}$ = 2.818, p = 0.039, permutation p = 0.035). The effects were very modest, with the haiku referencing color in insects containing only slightly more positive language in haiku than those that did not reference insect coloration (relative effect: positive tone = 0.52491). Research elsewhere suggests that colorful insects are less likely to elicit disgust in an observer, than their less colorful counterparts [37].

**Emotion over time.** To test whether there was detectable change in LIWC variables over time, we calculated Spearman's correlation coefficient. Our results reveal no correlations between these variables and time: positive tone ($\rho$ = 0.02086308, p = 0.2971); negative tone ($\rho$ = 0.000325706, p = 0.987); emotion ($\rho$ = -0.01276468, p = 0.5235); positive emotion ($\rho$ = 0.005387698, p = 0.7877); negative emotion ($\rho$ = -0.01772976, p = 0.3756; anxiety ($\rho$ = 0.02020234, p = 0.3126).

## Discussion

### Most commonly referenced taxa

**Araneae (spiders).** Spiders are frequently referenced in haiku (263 poems in the primary corpus, 220 in secondary corpus), but this is mostly a recent phenomenon. More than 98% of the references were published from 1960 onward. Taxonomically, there is a strong bias towards araneomorph spiders (72% of references in the secondary corpus), especially including those species that make orb and cob webs, when compared to the other major spider lineages: Mygalomorphae (3% of spider references) and Mesothelae (0%). About 25% of spider references could not be determined below Araneae. Active hunters, like the charismatic jumping spiders (Araneomorphae: Salticidae; 3% of spider references) and wolf spiders (Araneomorphae: Lycosidae; 1%) are hardly ever referenced. References of the unmistakable, diverse (1,047 spp.; [38]), and compelling tarantulas (Mygalomorphae: Theraphosidae), were scant in haiku (2% of spider references). The proportion of tarantula records in iNaturalist (<2% of spider records; [22]), however, is similarly sparse.

The most common biological attributes referenced in haiku about spiders relate to their extended phenotype, i.e., their webs (n = 131), and locomotion (n = 75). This result mirrors what we see in the AntConc analyses on the primary corpus. The top nine associated words from the cluster analysis of "spider", for example, are: web, webs, weaves, crawls, silk, jumping, moves, descends, and thread. Spiders were also more likely to be referenced as inside or around a home or building (domestic, n = 70; peridomestic, n = 23) than in nature (not domestic, n = 33), and they had the highest proportion of domestic references relative to the other arthropod orders analyzed. Interestingly, spiders comprise only 16% of all arthropod

taxa found indoors in a recent survey [39], and both abundance and diversity of spiders outdoors far exceeds those found indoors.

**Odonata (dragonflies, damselflies).**   Odonata were referenced in 182 haiku (135 in the secondary corpus), and, unlike for spiders, the references reveal no obvious historical bias. Culturally, they have been popular insects since the 1600s. References to Anisoptera (dragonflies) abound, encompassing 89% of all Odonata references. Only ∼5% of these references, however, included enough information to determine a taxon below suborder. Zygoptera (damselflies) makes up slightly more than half of all odonate species (3,218 of 6,330 described spp.; 50.8% [40]) and ∼32% of odonate iNaturalist records [22] and yet comprise a mere 10% of Odonata references in the secondary corpus. Anisoptera on average are larger, faster, and otherwise more conspicuous, which may account for the bias in their representation.

Flight (n = 54) was the most common biological trait referenced in haiku about Odonata, followed by references to color (n = 35). The cluster analysis of the primary corpus likewise yields terms related to flying (hover, hovers, flying, lands) and color (red, blue). Odonata also had the highest proportion of wild habitat references (not domestic, n = 48 or 75% of the habitat references for this taxon). Given the conspicuous nature of adult dragonflies and damselflies, with their bright colors and adept flight, life stage references were dominated by adults (122 references to imago vs. one for immature and three for egg). Immature odonates (called "naiads" [41]) live underwater and are camouflaged with their surroundings. They are not as readily observed as adults.

**Orthoptera (crickets, katydids, grasshoppers).**   "Cricket", "katydid", and "grasshopper" appear to be used interchangeably by translators of Japanese haiku [15, 42], which complicates our taxonomic assessment. Nevertheless, references to Orthoptera (308 haiku in the primary corpus; 204 in the secondary corpus) appear to be dominated by Ensifera, especially crickets (Gryllidae; 68% of orthopteran references). Ensifera accounts for ∼58% of all described orthopteran species [43] and ∼39% of orthopteran records on iNaturalist [22] but 79% of all Orthoptera references in the secondary corpus. Caelifera (grasshoppers and related orthopterans; ∼42% of Orthoptera spp., ∼60% of orthopteran records on iNaturalist) was represented by only 21% of the orthopteran references, most which were grasshoppers (Acrididea, probably mainly Acrididae). A recent survey of domestic environments found that the vast majority of home invaders are Ensifera (e.g., 100% of orthopteran home invaders in [39]), which may explain this bias in haiku. Ensifera also has more species that sing, compared to Caelifera, and these species tend to have louder calls than caeliferans [44]. Similar to Odonata, Orthoptera appear in haiku fairly consistently across time.

As expected, haiku that mention Orthoptera were dominated by references to sound production (n = 125, out of 204 haiku). Six of the top eight words revealed in the AntConc clusters include: chirp, chirps, song, cricky, sings, singing. These insects are well-known songsters and have been kept as companions by many cultures around the world [45, 46]. Other commonly referenced traits related to seasonality (n = 42) and time of day (n = 65). A defining feature of Orthoptera is their elongate, saltatorial hind legs, and yet we could only find eight references to jumping. Zero haiku referenced these insects' fighting behaviors, which otherwise is well represented culturally [45].

**Hemiptera (aphids, cicadas, true bugs).**   References to Hemiptera are fairly common and consistent across time in the primary corpus, with 274 references in haiku (192 in the secondary corpus). There is a bias towards Auchenorrhyncha (∼68% of Hemiptera references), driven by references to Cicadidae (>98% of Auchenorrhyncha references; 67% of all Hemiptera references). Cicadas, which tend to be large, conspicuous, and gregarious insects that sing loudly, are common cultural icons [47]. The next two most commonly referenced insects are water striders (Heteroptera: Gerridae), which skate along the surface of bodies of water,

and stink bugs (Heteroptera: Pentatomidae), also referred to as "fart bugs" by translators. Despite the extraordinary diversity of Sternorrhyncha (scale insects, whiteflies, aphids, etc.), which comprises >18,000 spp. [48] and is represented by >53,000 records in iNaturalist [22], and which includes species that are common pests but also species that produce important products (dyes, lac, manna) [49], these insects made up a paltry 7% of Hemiptera references. Almost all Sternorrhyncha references were about aphids (Aphidoidea).

As expected, given the popularity of Cicadidae, biological references were dominated by sound production (n = 89), with AntConc clusters including terms like chirr, chirrs, sing, sings, chorus, cries, cry, singing, and song. Other commonly referenced traits include their natural habitat (not domestic, n = 50) and allusions to their seasonality (n = 39) and time of day (n = 39).

**Hymenoptera (wasps, bees, ants).** References to Hymenoptera were found in 366 haiku in the primary corpus (347 in the secondary corpus) and were dominated by those species capable of stinging (Aculeata). Aculeates accounted for 96% of the references. For comparison, Aculeata includes 65,258 described species (42.7% of all Hymenoptera; [50]) and an astounding 1,241,890 records on iNaturalist (88.7% of all Hymenoptera records on that site; [22]). Bees (Anthophila; 43% of hymenopteran references) and ants (Formicidae; 42% of hymenopteran references), which together comprise more than 29,000 species, or about 20% of described Hymenoptera [50], are the most commonly referenced aculeates. As with spiders, haiku about Hymenoptera are mostly a recent phenomenon; only ∼6% of the references were published prior to 1959 (18/347 in secondary corpus). Given their ecosystem services (e.g., pollination), natural products (wax, honey), the long, global history of apiculture [51], the social nature of many species, and the prevalence of these insects in the environment (e.g., more than half the total biomass of terrestrial arthropods globally is ants [52, 53]) and in our homes [39], we expect to see many references to bees and ants in haiku. The paucity of references before 1959 is surprising.

Biological trait references for Hymenoptera mostly allude to foraging (n = 78) and related traits: pollination (n = 21), flying (n = 31), walking (n = 52). Top clusters from AntConc include words like scurry, walks, crawling, march, circles. Habitat was another commonly referenced trait: domestic (n = 31, primarily Formicidae), peridomestic (n = 47), not domestic (n = 71). Despite Aculeata accounting for 96% of the hymenopterans in haiku, poets made relatively few references to defense (n = 44) or to the sting itself (defensive structure, n = 5). Many aculeate species are social, and there were 74 references to sociality in the secondary corpus.

**Coleoptera (beetles, fireflies).** References to beetles (347 haiku in the primary corpus; 311 in secondary corpus) largely focused on species in Polyphaga (91%), especially fireflies (Lampyridae; 59% of Coleoptera references) and ladybird beetles (Coccinellidae; 18% of Coleoptera references). Given the cultural importance of these taxa [47, 54, 55], it is not surprising to see them represented frequently in haiku. However, Scarabaeoidea (e.g., scarabs, stag beetles) also includes many large, conspicuous, and culturally relevant beetles [56, 57]; yet they make up only about 9% of the beetle references in haiku. Another surprisingly underrepresented taxon is Cerambycidae, the longhorn beetles, which was referenced in only one haiku. These beetles are extraordinarily diverse (>30,000 species, ∼8% of all beetle species; [58]) and are represented by 210,413 records on iNaturalist (13.1% of beetle records and the second most reported beetle family; [22]). Carabidae likewise was represented in only two haiku, despite having about 40,000 species (>10% of all beetle species; [58]) and 126,483 records in iNaturalist (∼8% of all beetle records; [22]). References to Coleoptera across time are relatively consistent.

Bioluminescence (n = 111) was the most frequently referenced biological trait, which is unsurprising, given the popularity of fireflies in haiku. Other common traits relate to the behavior of flashing of fireflies, especially night references (temporal, n = 86), phenology

(n = 38), and flight (n = 55). The top 20 AntConc clusters include terms related to locomotion (flits, flitting, come, flies), phenology (first), and bioluminescence (calling, lights).

**Diptera (mosquitoes, midges, flies, maggots).** We found 197 references in the primary corpus but 314 in the secondary corpus, where determinations were more confidently assigned. The word "fly" returned numerous irrelevant results in AntConc (flying, butterfly, firefly, etc.) Diptera references were fairly evenly distributed over time. Taxonomically, we see an high representation of mosquitoes (Culicidae; 42% of Diptera references), relative to other families. For context, mosquitoes comprise 3,725 species ($\sim$2.4% of all Diptera; [59]) and 18,205 records in iNaturalist ($\sim$2.8% of all Diptera records; [22]). Mosquitoes are abundant and often pestiferous in the adult stage, with many species capable of vectoring important human and veterinary diseases. Other biting flies, including horse flies (Tabanidae), black flies (Simuliidae), and tsetse flies (Glossinidae), together received as many references as the common house fly (Muscidae: *Musca*). At least 39% of Diptera references were not determinable below the order.

The polysemic nature of the word "fly" complicated our AntConc cluster analysis, but we did find many words associated with mosquito behavior (swarms, swarming, whine, whines, pesky) and cultural control (net, nets, and "smudge", which refers to burning substances that produce a repellent smoke). Of the biological traits scored in the secondary corpus, ecological variables were by far the richest, especially habitat (n = 167), phenology (n = 55), and temporal (n = 46) references. Behavior was the second richest category, especially traits related to feeding (foraging, n = 18; parasitism, n = 38) and sound production (n = 25).

**Lepidoptera (butterflies, moths).** Lepidoptera are the most commonly referenced insects in haiku, with 498 references in the primary corpus and 510 references in the secondary corpus. About 68% of lepidopteran references are about butterflies (Papilionoidea). Monarchs, *Danaus plexippus* (Linnaeus, 1758), and cabbage whites, *Pieris rapae* (Linnaeus, 1758), were the most identifiable species in the secondary corpus. These species are abundant, well-known (e.g., see [60]), and widespread, with 156,878 and 61,613 records, respectively, on iNaturalist [22]. The relative commonness of bagworm references (Psychidae; 1,350 spp., <1% of all Lepidoptera spp.; [61], compared to other small families, was surprising to us but mirrors references to this taxon in other cultural contexts [62–64]. Lepidoptera references were distributed fairly evenly across time.

As with most other arthropod references, the adult stage is the primary focus (imago, n = 388). Lepidoptera did have the most references to immature stages (n = 92), though, given the familiarity of caterpillars. The most frequently referenced traits overall were flight (n = 161), color/pattern (n = 92), and habitat (not domestic, n = 121, peridomestic, n = 65; domestic, n = 40). The top 27 AntConc clusters include terms related mainly to size (little), locomotion (flits, flitting, lands, dance, fly), color (white, blue, yellow, black), and to the type of lepidopteran (tiger, monarch, luna).

## General impressions

Altogether, the references to biological traits mirror what is generally understood about children's knowledge of insect biology and which aspects of insect biology we suspected would resonate with members of the general public. Children readily recall details about an insect's ecology [65] and feeding behavior [66], for example, but have a poor understanding about how insects reproduce [67] and about arthropod anatomy beyond the presence of wings, legs, and an exoskeleton [68, 69]. We found no equivalent knowledge surveys for adults, who comprise the vast majority of poets in our data set, but it is possible that these trends hold true in the haiku-writing public, independent of age.

Regardless, our results provide insights into which arthropods and biological traits are notable (or not) to people. This knowledge provides entomologists and educators with target subjects for broader engagement. For example, aquatic arthropods provide an extraordinary array of ecosystem services [70], serve as critical bioindicators [71] and sources for bioinspiration [72], and they are commonly collected as bait. Haiku about these arthropods and their biology, however, are surprisingly rare. For example, not a single poem in our primary corpus mentioned Trichoptera (caddisflies, 14,999 species), Plecoptera (stoneflies, 3,788 species), nor Megaloptera (alderflies, dobsonflies, 354 species). A single poem referred to Chironomidae, which, with >7,300 species, is the most diverse family of aquatic insects [73]. These taxa represent a stunningly diverse array of compelling life history traits, and many of the species are common, conspicuous, and charismatic. The absence of aquatic arthropods in haiku may represent a gap in the public's knowledge of these organisms and hence an opportunity for novel engagement.

Alternatively, entomologists could use the results of these analyses to strengthen their outreach with more references to the known. Reinforcing programs and curricula with well-understood taxa and traits—butterflies and their colors, spiders and their webs, fireflies and bioluminescence, etc.—might allow for more efficient transfer of knowledge about arthropods.

We also recognize some limitations of our study. First, we intended our results to reveal, in part, something about the human emotions related to certain taxa and traits and how they may have changed over time. Our analyses of the secondary corpus, however, revealed very few obvious trends regarding emotion and tone, probably because haiku are short ($\bar{x} = 9.8176$ words) and intentionally subtle. The reader is meant to infer the emotion. Additionally, applying statistics to questions related to language is inherently challenging because language is essentially non-random [74]. However, our analyses were performed on the scores generated by LIWC analyses, rather than on the language itself. The results are accompanied by the relative effects when significance was found, which provide context for the magnitude of effect found from the statistical tests.

Second, our corpus is quite uneven in its representation of haiku across time (S4 Fig), poet type, and geographic region. Amateur writers, especially children, might be more likely to incorporate words that are represented in the LIWC-22 word banks (i.e., to use language that is more overtly emotional). The poems by children, however, are represented in our data set only from 2008–2012. Interestingly, the strongest emotional trend revealed in these poems relate to the negativity surrounding scorpions. Most of the scorpion haiku (n = 14/15) are from the Hexapod Haiku Challenge submissions, and they were largely written by children under the age of 13 (n = 12/15). Likewise, poets in certain geographic regions may limit their representation of insects to those taxa that are readily observed in those regions and/or those taxa that are traditionally included in the medium (i.e., literary inertia). For example, the poems in our corpus that were written prior to the 1920s are exclusively from Japan, whereas the poems from after that decade are increasingly international.

## Conclusion

Computational lexicology [75] and related quantitative analyses of culture [76] and knowledge [77] continue to reveal insights that potentially inform education, outreach, and conservation. Poetry stands as a compelling medium to further prospect for these insights, given the deliberative word choice and frequent inclusion of nature. Haiku, which are poems that intentionally and frequently represent moments in nature, are especially relevant in this context. Our analyses of nearly 4,000 representative poems (more than 37,000 words) reveal many surprising results, articulated above, that have already catalyzed conversations locally about future

engagement activities. Studies of larger, more even bodies of text would undoubtedly allow for broader and more robust conclusions about how different arthropods are perceived by the general public and whether outreach programs can be further adapted for more effective engagement.

## Supporting information

**S1 Table. Arthropod representation in haiku.** Columns represent (L to R): taxon, including common names; number of described species in each taxon [78, 79]; percent of total relevant arthropod species; number of occurrences for each taxon in iNaturalist [22]; percent of relevant arthropod occurrences in iNaturalist; number of references in the 2500 haiku corpus for each taxon; percent of all arthropod references in the corpus. Rows in **bold** had zero references in haiku. Rows highlighted in yellow are taxa with few species and no references in haiku. Rows in orange represent taxa that are surprisingly underrepresented.
(PDF)

**S1 File. Secondary corpus with results.** This spreadsheet contains all the haiku analyzed as the secondary corpus, with results from the trait scoring and the LIWC-22 analysis.
(XLSX)

**S2 File. Primary corpus.** This spreadsheet contains all the haiku analyzed as the primary corpus, which includes the haiku in the secondary corpus. This corpus was used with AntConc, to get lists of words associated with arthropods.
(XLSX)

**S1 Fig. Taxonomic representation through time.** This figure illustrates the proportion of haiku that reference any particular taxon at a particular period of time.
(TIF)

**S2 Fig. Trait category representation through time.** This figure illustrates the proportion of haiku that reference the major trait categories in each decade.
(TIF)

**S3 Fig. Top 20 traits through time.** This figure illustrates the proportion of haiku that reference the top 20 traits in each decade.
(TIF)

**S4 Fig. Number of haiku in the primary corpus over time.** This histogram illustrates the number of poems per year that are represented in the primary corpus.
(TIF)

**S1 Appendix. Corpus construction.** This document describes the strategies and resources used in constructing the corpora of haiku that reference arthropods. This document is also available through Penn State's institutional repository, ScholarSphere: https://doi.org/10.26207/gpxg-q347.
(PDF)

**S2 Appendix. Biological variables and scoring methods.** This document describes the process used to score haiku for biological complexity and taxon. This document is also available through Penn State's institutional repository, ScholarSphere: https://doi.org/10.26207/04h1-a695.
(PDF)

**S3 Appendix. AntConc analyses.** This document describes the cluster analyses and results using the AntConc application. This document is also available through Penn State's institutional repository, ScholarSphere: https://doi.org/10.26207/35sv-ep54.
(PDF)

## Acknowledgments

We thank Heather Froehlich (University of Arizona), for her invaluable assistance throughout the project; Anne Burgevin, for inspiring us to further study haiku; Hojun Song (Texas A&M University), for his input regarding Orthoptera diversity and biology; and David Lanoue (Xavier University) for allowing us to include his translations of Issa haiku in this research. We also appreciate the insights provided by our anonymous reviewers, comments and questions greatly improve the manuscript.

## Author Contributions

**Conceptualization:** Andrew R. Deans, Laura Porturas.

**Data curation:** Andrew R. Deans, Laura Porturas.

**Formal analysis:** Andrew R. Deans, Laura Porturas.

**Methodology:** Andrew R. Deans.

**Project administration:** Andrew R. Deans.

**Visualization:** Andrew R. Deans, Laura Porturas.

**Writing – original draft:** Andrew R. Deans, Laura Porturas.

**Writing – review & editing:** Andrew R. Deans, Laura Porturas.

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
