## [Decision Letter · Decision Letter 0]

7 Dec 2023

PONE-D-23-33498Diversity and complexity of arthropod references in haikuPLOS ONE

Dear Dr. Deans,

Thank you for submitting your manuscript to PLOS ONE. After careful consideration, we feel that it has merit but does not fully meet PLOS ONE’s publication criteria as it currently stands. As an editor, I have to say that your article is quite different from the "standard experimental" manuscripts I am used to edit, nevertheless this is a very interesting and pleasurable topic. It is certainly also important, as awareness of insects (or nature in general) is key to promote nature conservation issues. As reviewers, I asked two leading and enthusiastic entomologists for their opinion. One is an experienced popular science writer with a faible for the arts and history, the other is an expert in insect behaviour and is familiar with Japanese culture. Both find your study very interesting and worthwile, and provided insightful and valuable comments that should be considered.  Therefore, we invite you to submit a revised version of the manuscript that addresses the points raised during the review process.

We look forward to receiving your revised manuscript.

Kind regards,

Christian Wegener

Academic Editor

PLOS ONE

Journal Requirements:

Additional Editor Comments:

I was not familiar to the topic, and only had a vague idea what a Haiku is. Your introduction certainly helped me to understand better, but it is very short, and gives references mostly to books that are hard to get for a biologist/entomologist. By a publication search I found a very helpful report by R.R. Dunn on insects in traditional Japanese Haiku, that helped me much to better understand why insects occur in these poems.

You may not find it required, but I suggest to reference this article which is available in a journal readily accessible to entomologists:

 Dunn RR. Poetic Entomology: Insects in Japanese Haiku. American Entomologist. 2000;46: 70–72. doi:10.1093/ae/46.2.70  

Reviewers' comments:

Reviewer's Responses to Questions

**Comments to the Author**

1. Is the manuscript technically sound, and do the data support the conclusions?

Reviewer #1: Partly

Reviewer #2: Yes

2. Has the statistical analysis been performed appropriately and rigorously? 

Reviewer #1: No

Reviewer #2: Yes

3. Have the authors made all data underlying the findings in their manuscript fully available?

Reviewer #1: Yes

Reviewer #2: Yes

4. Is the manuscript presented in an intelligible fashion and written in standard English?

Reviewer #1: Yes

Reviewer #2: Yes

5. Review Comments to the Author

Reviewer #1: The manuscript by Deans & Porturas represents a very interesting case of cultural entomology and should be appealing to a wide audience within and outside entomology and cultural history research. The analysis is extensive and thorough and the data basis, and database, are rich. It certainly is a unique study. I really liked and engaged with it – and possibly for that reason I have some criticisms. Some can, other probably cannot be addressed, others the authors may not wish to address. Here they are anyway, I hope the authors find them useful:

1) I feel that there is a distinct N American, possibly, US-limited angle of the study, which in case of a traditional Japanese poem type is somehow a little ‘odd’. This concerns the overrepresentation of haiku in the English language, the overrepresentation of US school kids and also the few parts where the Discussion discusses something, also mainly findings from the US (460-465). One suggestion here would be to be a bit more open about the geographic origin of the haiku. Perhaps also about how many are from children (see L. 498).

A broader issue here has been briefly alluded to in the discussion (L 333-335) – but if I see haiku about barklice, proturans or diplurans I would imagine the translation be quite imprecise. My guess is that not even all entomologists would straight away know what these are, not to say, how to translate them?

2) I did not understand what criteria led to the generation of the 2500 haiku subset, other than that it includes every author.

3) I have not read many haiku but from other fiction literature it is quite clear that many authors/ poets know that cricket or cicada singing, or firefly glowing, actually are for mate attraction – and so I could imagine that various poets knew they were writing about reproduction?

Swarming was not a behavior term – did swarming not occur in haiku? It might also be reproduction.

I did not understand what ‘association’ means under variable “ecology!

4) The statistical analysis I do not fully understand. The authors use an lm approach (L 129) and then some kind of transformations if the data are not parametric (which they aren’t). Why are not simple glm approaches used where the variance structure can be specified through ‘family’ and ‘link’? This would allow to properly control for the skewed variances? But it may just be me and glm does not work on multivariate data sets. However, “he number of unique orders or the average biological complexity score” (L 130) are not multivariate.

Personally, I would use, or additionally use, a glmm approach with insect order as a random effect – so all parameters analysed over time would be examined within individuals orders. But maybe that’s an overkill.

On line 150, it was said that haiku were excluded... – how many of such haiku were excluded. The results still speak of 2500 haiku? So it is not clear to me when what was excluded, and how many.

In L. 227 haiku type is used as a predictor but it is not explained what haiku ‘type’ is.

5) I personally don’t really need it (I find the analysis interesting without explicit explanation) but the feeling crept in that the purpose of the study wasn’t entirely clear. If the purpose was to just characterise the diversity in haiku, then why was inaturalist chosen as a comparison (I like the idea but maybe spell out, why). I was also reminded of the two careful studies of insect representation on paintings by Dicke – I use them in my lectures: Dicke M 2000. Am. Entomol, Dicke 2004 Proc Neth Ent Soc).

If, however, the specifics of haiku were to be studied, then one would possibly expect a comparison with other fiction literature.

6) The time axis. I think it would be great to have at least one figure with the time axis in the main ms, S1 or S2.

If there are 2500 haiku over 400 years (avg 6/year) – some years will come out with a very low number of haiku to calculate diversity. Would it be useful to restrict the trend analyses to years with, for example, 10+ haiku? Also for L 202-204?

The main problem, I think, is that any temporal trends are distorted by the large abundance of recent haiku. Calling them ‘somewhat uneven’ (L. 494) is daring.

7) Fig 1. Compares species diversity of haiku with inaturalist. However, inaturalist is only explained later. Also, it should be in the Methods, not the Results. Why was inaturalist chosen, not , say flickr which seems a more general audience? The source of the species numbers per order is not given, nor is it mentioned that (presumably) the global species number is given.

8) For the emotional trends we would need some overall estimate of how many were positive and negative, otherwise the relative importance of the two categories is not clear in any comparison.

Moreover, in what the authors call post hoc hypotheses, in all three it is stated the frequent use is associated with alleged positive emotions.

9) The authors erect some ‘hypotheses’. I think the authors are under a misconception. A hypothesis is a directional statement that is derived from theory or data. It is not a loosely noted speculation without any justification. I think the comparisons are fine but pressing them into a hypotheses corset is incorrect. I recommend the editor to insist on addressing this point, or to provide a proper derivation why adults should be preferred over larvae etc.

My personal suggestion is to kick them out but use exactly those consideration of adults vs larvae, indoors vs outdoors etc in the Discussion.

10) ‘hypothesis’ 2 argues with increasing urbanisation. That may be so but in any case, the resulting prediction would a change in the temporal trend that follows the alleged urbanisation, not an overall, historically unchanged %.

11) Most of the Discussion is Results, specific for individuals orders. For example in spiders, only the last two lines are a discussion while, to the contrary, all the lines before ‘discuss’ results that are not mentioned in the Results, including the sample sizes. I suggest to disentangle this for all orders. Whether the details stay in here or are in an appendix, I don’t want to suggest. However the precise numbers for inaturalist or the global species database seem appendix candidates, the % for the main text????

12) Smaller aspects.

L 195. Lower recent references to fleas are explained with improved pest control. This connection seems questionable to me – fleas are possibly among the last animals that pest controlelrs are bing called for?

L. 197. Bedbugs have resurged globally since 1980, and especially since the 2000’s. However, given the large amount of references to bedbugs throughout history (ref 29, also see ‘Bedbug’ from Reaktion Books London) I would suggest that the suggested cause and effect is a mis-citation. The restriction to contemporary mentions of bedbugs might be entirely related to the distorted sample size in this area.

L. 210-211. Would there be room to display the haiku with the highest and the lowest score?

L. 484. Some of the aspects mentioned here are not limitations of the study, they simply are negative, or neutral effects. Limitations are aspects that introduce biases.

Reviewer #2: This paper deals with the following topics; 1) What kind of arthropod is chosen in haiku? Representation degree of each arthropod in haiku was evaluated as over, under or equal, in view of their species diversity and naturalist observation. 2) What trait or aspect of the arthropod in haiku attracted poet's ? 3) Where did poets meet the arthropod referred, that is, indoor or outdoor ? 4) What did the poets feel for the arthropod concerned, positive or negative? 5) Have the answers to the above topics changed over time ?

The results indicated that most of the answers to the above questions were by and large what were easily expectable with a few exceptions. Almost no difference was found for time. The methods are solid, the results were clearly presented and the discussion is modest. This paper is unique with respect to the use of the reference in haiku to estimate the public concern with arthropods and nature. In Japan, not a few books on arthropods in haiku have been published. The most frequently referred 8 orders of the arthropods in Japanese books are the same as the top 8 orders in this paper, indicating that the results and analysis is reliable. Information obtained is useful for nature education at school and effective engagement in conservation activity in the public as the authors stated.

The authors described that poems were written prior to 1920s are exclusively from Japan, whereas the poems from after the decade are increasingly international. For the latter, I recommend that names of the countries referenced and the number of haiku sampled in each country are attached to the method section, if possible, because biological diversity, nature and culture are different between Japan and other countries.

Finally, I have special interest in one of the authors' statements: The relative commonness of bagworm references, compared to other small families, was surprising to us but mirrors references to this taxon in other cultural context. What is other cultural context ? Explain this more concretely.

6. PLOS authors have the option to publish the peer review history of their article (what does this mean?). If published, this will include your full peer review and any attached files.

Reviewer #1: No

Reviewer #2: No

---

## [Author Response · Author response to Decision Letter 0]

19 Jan 2024

Reviewer #1

(1) "I feel that there is a distinct N American, possibly, US-limited angle of the study, which in case of a traditional Japanese poem type is somehow a little `odd'. This concerns the overrepresentation of haiku in the English language, the overrepresentation of US school kids ..."

Deans and Porturas: One of our regrets in developing the corpus initially was that we did not include country of origin for many of our poems. We have those data for only 2,518 of the poems in the primary corpus (out of 3,894 poems total). Of those, 1,706 are from poets outside of the US and Canada, including more than 1,474 from Japan (mainly by Kobayashi Issa). If I had to guess, I'd say that about 1,800 or 46% of the poems in our primary corpus are from North America. Haiku is now a global phenomenon, and we made a concerted effort to gather haiku from across Africa (n=73, not included in the above numbers), the Indian subcontinent, eastern Europe, Australia, New Zealand, and elsewhere. Given that we had no funding for this project we had to limit our haiku harvest to resources that were (mostly) available via the Web and which were available in the language we understand.

Regarding age, it's true that the haiku by children are dominated by students from the US. Haiku by poets under 13 number only ~250 in the primary corpus, and probably 75% of those are from kids across the US. The vast majority (93.6%) of poems in our data set, however, were written by adults. The points made in our discussion (lines 460--465) are about haiku in general, not just haiku by kids. We added a sentence in the materials and methods to clarify the composition of our source material.

It would be difficult to rectify these issues in the current manuscript, as it took us more than six months just to assemble the primary corpus and another six months to do the analyses. We do, however, have a growing set of additional (and increasingly international) haiku that were not included in these analyses. In follow-up research, especially with collaborators in Japan and in expanding to other types of poetry, we will refine our approach to get more robust answers to these questions. 

(2) "I did not understand what criteria led to the generation of the 2500 haiku subset, other than that it includes every author."

Deans and Porturas: Scoring each poem for 69 variables was labor intensive, and we simply did not have the time nor the resources to continue the process for each poem in the corpus. We stopped scoring biological traits after 2500 haiku, which we felt was a sufficient and representative dataset. We included the haiku that we had compiled but not scored for biological traits in the linguistic analyses (e.g. AntConc word clusters), because they allowed us to look for broader trends without much extra effort. 

(3) "... from other fiction literature it is quite clear that many authors/ poets know that cricket or cicada singing, or firefly glowing, actually are for mate attraction – and so I could imagine that various poets knew they were writing about reproduction?"

Deans and Porturas: In most cases, haiku, which only have about 10 words, did not have enough content to make it obvious that the poet recognized these behaviors as courtship/reproduction. Reviewer #1's comment here does point to a larger challenge we had when scoring biological traits in text that is literary, rather than scientific. Sometimes our scores reflect an "inferred} biology, rather than an explicit one. If we could comfortably infer that the author recognized the biology, we scored it at +1. For example, ``Sadly I see /

the light fade on my palm: / a firefly'' would be scored as +1 for bioluminescence and +1 for physiology, but ``firefly love / glowing brightly / for just a moment'' would be scored as +1 for bioluminescence, +1 for physiology, +1 for courtship, and +1 for reproduction.

"Swarming was not a behavior term – did swarming not occur in haiku? It might also be reproduction."

Deans and Porturas: Very few haiku referred to ``swarming'' (n=15). A ``swarm'' of any taxon received +1 for the broadest category of ``behavior'', and if we could comfortably infer more specific biology, that also received a score. A swarm of bees, for example, would be scored +1 for sociality and +1 for behavior, while a swarm of flies over a dead carcass would be +1 for foraging and +1 for behavior. A ``swarm'' of dragonflies would simply be scored as +1 for the broadest category of ``behavior''.

"I did not understand what `association' means under variable ``ecology''!"

Deans and Porturas: Our supplementary methods file defines these variables more explicitly. Association is ``inter-species interaction, excluding parasitism and predation'', and the example we provide as guidance is ``Ants herd aphids ...''. The ants in this case are associated with aphids. The Darwin Core term associatedTaxa was our inspiration: \\url{https://dwc.tdwg.org/list/#dwc_associatedTaxa}

(4) "The statistical analysis I do not fully understand. The authors use an lm approach (L 129) and then some kind of transformations if the data are not parametric (which they aren't). ... But it may just be me and glm does not work on multivariate data sets. However, ``The number of unique orders or the average biological complexity score'' (L 130) are not multivariate ... In L. 227 haiku type is used as a predictor but it is not explained what haiku ‘type’ is."

Deans and Porturas: Thank you for for this comment. We now see that the description of statistics was unclear. We used three separate analyses: 

A. The lm approach to determine whether time, haiku source, or # of authors was a predictor for (a) taxon diversity (# of unique orders/year) & (b) average biological complexity score. 

B. The nonparametric multivariate analyses were used to determine whether any of the LIWC variables related to the tone and emotion of haiku in the dataset (tone pos, tone neg, emotion, emo pos, emo neg, emo anx) were dependent on the categorical groups we were interested in (e.g., adult vs. immature, or domestic vs. not domestic).

C. Spearman's correlation tests were used to determine whether there were any correlations between the LIWC variables and time (though it seems that this was not part of the initial confusion).

We agree that glm approaches are often be more appropriate than linear modeling. We have followed your advice and used glm approaches to test whether time, haiku source, or # of authors was a predictor for taxon diversity referenced in haiku, and used a quasi-poisson regression because data was discrete and overdispersed. We also took your advice to add order in as a random effect when determining if time, haiku source, or # of authors was a predictor for biological complexity referenced in haiku. We used a linear mixed effects model for this because data was normal and continuous so Gaussian distributions seemed appropriate for analysis. For both of the above, we also restricted the analyses to years with 10 or more haiku (the new analyses only slightly changed interpretation). However, we did leave the multivariate analyses as-is. We recognize that nonparametric analyses are less powerful, but it is our impression that nonparametric tests are more appropriate when the dependent variable is nominal. We have modified the methods and results to reflect the new analyses, and have tried to clarify our descriptions of the statistics used. 

(5) "... the feeling crept in that the purpose of the study wasn't entirely clear. If the purpose was to just characterise the diversity in haiku, then why was inaturalist chosen as a comparison (I like the idea but maybe spell out, why). ... If, however, the specifics of haiku were to be studied, then one would possibly expect a comparison with other fiction literature."

Deans and Porturas: In our introduction we state that ``Collectively, [haiku] offer opportunities to understand how and which organisms inspire people, which biological traits resonate with the human experience, and where openings may exist for more effective outreach regarding biodiversity and conservation.'' We are also interested in seeing whether declines in insect diversity, which are now all over the popular press, are reflected in cultural products, like poetry. (I think maybe it's too early to see these effects.) We return to these themes in the General Impressions section and Conclusion---e.g., ``Computational lexicology ... and related quantitative analyses of culture ... and knowledge ... continue to reveal insights that potentially inform education, outreach, and conservation.''---with some examples from our results (e.g., focusing education and outreach on aquatic insects, which don't often appear in haiku). Hopefully that's clear enough for most readers. Comparisons across literary styles and other cultural products are beyond the scope of this work but certainly would be compelling.

(6) "The time axis ... The main problem, I think, is that any temporal trends are distorted by the large abundance of recent haiku. Calling them `somewhat uneven' (L. 494) is daring."

Deans and Porturas: Fair enough! We changed ``somewhat'' to ``quite''. We also restricted the trend analyses to years that had 10 or more haiku as recommended.

(6) "Fig 1. Compares species diversity of haiku with inaturalist. However, inaturalist is only explained later. Also, it should be in the Methods, not the Results. Why was inaturalist chosen, not, say flickr which seems a more general audience? The source of the species numbers per order is not given, nor is it mentioned that (presumably) the global species number is given."

Deans and Porturas: iNaturalist has >3 million registered users, many of which (most?) are amateurs. More importantly, though, the app has a huge community of competent identifiers and robust mechanisms for querying their data, using a taxonomic hierarchy. The vast majority of results are relevant to our questions, and contaminants (e.g., copyrighted images) are rapidly flagged and removed by the community. We added a few sentences regarding iNaturalist to the Methods, as recommended by the reviewer. 

Flickr is an interesting alternative and maybe(?) provides and even more amateur perspective on which arthropods humans find interesting. As an extensive Flickr user myself, I find it much more difficult to extract similarly robust data, though. For example, a query of ``lepidoptera'' yields almost 380k images, whereas a query of ``butterfly'' yields almost 990k images. A closer look at the butterfly results yields further reveals many images that are confounding (tattoos, knives, women in bathing suits) or otherwise not representative of human-arthropod encounters (1,747 book plates, uploaded by the Biodiversity Heritage Library). 

The sources for species number are indeed cited, but the reader had to dig for those references in the Discussion. We edited the figure caption to make this a bit clearer.

(8) "For the emotional trends we would need some overall estimate of how many were positive and negative, otherwise the relative importance of the two categories is not clear in any comparison. Moreover, in what the authors call post hoc hypotheses, in all three it is stated the frequent use is associated with alleged positive emotions."

Deans and Porturas: We are not sure we fully understand what the reviewer is suggesting, and think the confusion is because of our unclear descriptions from the first submission. For the emotional trends, we are examining whether the biological categories of interest that we have assigned haiku to (e.g., adult vs. immature, domestic vs. not domestic, different taxonomic orders) influence the language used by authors in haiku. This is measured by the scores for the different linguistic categories generated by LIWC analyses (tone pos, tone neg, emotion, emo pos, emo neg, emo anx). These scores are the proportion of words in the haiku that match to the LIWC software word banks for each of the different linguistic categories/variables. If a haiku has 9 words total and 3 of them match words in the tone\\_pos word bank, the haiku would have a score of (3/9 =) 0.333 for that variable. If it also has 1 word that matches words in the tone\\_neg word bank, it receives a (1/9 =) 0.111 for that variable. Since scores are proportions, they are all positive between 0--1. We don’t have count measure of how many haiku from each biological category of interest were ``positive'' or ``negative''. We do, however, have a measure of relative effect (probability that a random sample from one group is higher than a random sample from the other group). These are reported when there are differences found between biological categories. We did not think it necessary to report these effects if no significant differences between biological categories are found.

We revised the emotional trends section of the results and tried to add clarity about this in the methods section.

(9) "... A hypothesis is a directional statement that is derived from theory or data. It is not a loosely noted speculation without any justification. I think the comparisons are fine but pressing them into a hypotheses corset is incorrect ..."

Deans and Porturas: We agree and acknowledge our ``hypotheses'' in the emotional trends section were primarily based on impressions we have from communications with visitors to our museum and from some published papers, rather than being derived from any theory. We have kept the comparisons, because we do feel as though they might be of interest to readers, but removed any language that suggests comparisons were driven by hypotheses. 

(10) "`hypothesis' 2 argues with increasing urbanisation. That may be so but in any case, the resulting prediction would a change in the temporal trend that follows the alleged urbanisation, not an overall, historically unchanged %."

Deans and Porturas: We removed this part of the results section.

(11) "Most of the Discussion is Results, specific for individuals orders. For example in spiders, only the last two lines are a discussion while, to the contrary, all the lines before ‘discuss’ results that are not mentioned in the Results, including the sample sizes. I suggest to disentangle this for all orders. Whether the details stay in here or are in an appendix, I don’t want to suggest. However the precise numbers for inaturalist or the global species database seem appendix candidates, the % for the main text?."

Deans and Porturas: This situation is likely an artifact of an earlier draft, where the results and discussion were merged. We struggled to completely disentangle these elements for this manuscript, hence the addition of this sentence in the results: ``These trends are discussed further in the taxon-specific summaries below.'' We edited this section a bit, to make it clearer.

(12) "L 195. Lower recent references to fleas are explained with improved pest control. This connection seems questionable to me – fleas are possibly among the last animals that pest controllers are being called for?"

Deans and Porturas: It's not true that fleas are ``among last animals'' that are treated by professional pest controllers, but indeed, there seems to be less of a problem with fleas in contemporary society. The reduction of fleas as common pests is almost definitely a multifaceted phenomenon, due in part to rodent control, increased urbanization (we added a citation to support this) and to increased control involving companion animals and the application of imidacloprid and fipronil spot treatments and lufenuron (we added a reference here as well). We know of no analogs with respect to bedbugs or lice. In fact, increasing urbanization may facilitate bedbug problems. We added some language to manuscript to make this clearer.

"L. 197. Bedbugs have resurged globally since 1980, and especially since the 2000's. However, given the large amount of references to bedbugs throughout history (ref 29, also see `Bedbug' from Reaktion Books London) I would suggest that the suggested cause and effect is a mis-citation. The restriction to contemporary mentions of bedbugs might be entirely related to the distorted sample size in this area."

Deans and Porturas: True. We added some language to this section to acknowledge more clearly the unevenness of our sampling and the possibility that translation errors may influence these results.

"L. 210-211. Would there be room to display the haiku with the highest and the lowest score?"

Deans and Porturas: We added to the text the one haiku with the highest score and an example of a haiku with a score of zero.

Reviewer #2

(1) "The authors described that poems were written prior to 1920s are exclusively from Japan, whereas the poems from after the decade are increasingly international. For the latter, I recommend that names of the countries referenced and the number of haiku sampled in each country are attached to the method section, if possible, because biological diversity, nature and culture are different between Japan and other countries."

Deans and Porturas: Reviewer #1 also picked up on this limitation of our study. In the current manuscript it will be difficult to rectify this bias and even to record country of origin for all poems. Poems in an Irish journal of haiku, for example, may have been written by poets in Canada, the US, or even Romania or Australia. Most of time this information is not included in the journal and must be pieced together by finding other poems, presumably by the same poets, in other journals that do report country of origin. This shortcoming is something we hope to rectify in follow-up studies.

(2) "I have special interest in one of the authors' statements: The relative commonness of bagworm references, compared to other small families, was surprising to us but mirrors references to this taxon in other cultural context. What is other cultural context ? Explain this more concretely."

Deans and Porturas: The reference associated with this statement mentions that there is a bagworm Pokemon (Burmy, Wormadam, and maybe Pineco). We add two more references to this statement, incuding one that describes the cultural relevance of bagworms in sub-Saharan Africa, for example as food and as sources of mythology.

---

## [Decision Letter · Decision Letter 1]

1 Feb 2024

Diversity and complexity of arthropod references in haiku

PONE-D-23-33498R1

Dear Dr. Deans,

We’re pleased to inform you that your manuscript has been judged scientifically suitable for publication and will be formally accepted for publication once it meets all outstanding technical requirements.

I would like to point out that rev#1 makes a reasonable comment to include parts of your answers in your rebuttal letter. I fully agree with his notion, it would make data use more transparent for all readers. If you decide to add a few words on that, the manuscript will not be re-reviewed and will not be delayed, and you will be provided a possibility to upoad the revised version, likely with the email detailing the required amendments.

Thanks for choosing PLOS ONE for publishing your quite unique and interesting work -

Kind regards,

Christian Wegener

Academic Editor

PLOS ONE

Reviewers' comments:

Reviewer's Responses to Questions

**Comments to the Author**

1. If the authors have adequately addressed your comments raised in a previous round of review and you feel that this manuscript is now acceptable for publication, you may indicate that here to bypass the “Comments to the Author” section, enter your conflict of interest statement in the “Confidential to Editor” section, and submit your "Accept" recommendation.

Reviewer #1: (No Response)

Reviewer #2: All comments have been addressed

2. Is the manuscript technically sound, and do the data support the conclusions?

Reviewer #1: Yes

Reviewer #2: Yes

3. Has the statistical analysis been performed appropriately and rigorously? 

Reviewer #1: Yes

Reviewer #2: Yes

4. Have the authors made all data underlying the findings in their manuscript fully available?

Reviewer #1: Yes

Reviewer #2: Yes

5. Is the manuscript presented in an intelligible fashion and written in standard English?

Reviewer #1: Yes

Reviewer #2: Yes

6. Review Comments to the Author

Reviewer #1: I think the authors did a great job in responding, explaining, polishing, changing, re-analysing etc. Some of the authors’ responses appear – to me – suitable to be put in the ms, to explain to the readers, rather than to the Editor and reviewers?

(Examples:

I would imagine it doesn’t do harm if in the paper it would also be stated that for logistical reasons the first 2500 haiku were scored, not all? So other people do not start wondering about the same thing as I did.

Or the justification why inaturalist, not... flickr or whatever.

Or the statement that nearly half of the haiku were from the US)

After doing the review I found two references that I think might be relevant, especially the first. I am not suggesting the authors should cite them, but I’d like to share them anyway.

DOI: 10.1002/pan3.10256

DOI: 10.1002/pan3.10551

Good luck, it is a fine and unique contribution

Reviewer #2: I hope authors continue their reseach and publish more comprehensive and analytical work of insects in haiku, with multilateral viewpoint such as nationality, natural features, culture, art and literature and tradition.

7. PLOS authors have the option to publish the peer review history of their article (what does this mean?). If published, this will include your full peer review and any attached files.

Reviewer #1: No

Reviewer #2: No

---

## [Editor Report · Acceptance letter]

7 Mar 2024

PONE-D-23-33498R1 

PLOS ONE

Dear Dr. Deans, 

I'm pleased to inform you that your manuscript has been deemed suitable for publication in PLOS ONE. Congratulations! Your manuscript is now being handed over to our production team.

Kind regards, 

on behalf of

Prof. Dr. Christian Wegener 

Academic Editor

PLOS ONE